# PeerJ

# ABT-737, a small molecule Bcl-2/Bcl-xL antagonist, increases antimitotic-mediated apoptosis in human prostate cancer cells

Ricardo Parrondo[1], Alicia de las Pozas[1], Teresita Reiner[1] and Carlos Perez-Stable[1,2,3]

[1] Geriatric Research, Education, and Clinical Center and Research Service, Bruce W. Carter Veterans Affairs Medical Center, Miami, FL, USA
[2] Division of Gerontology & Geriatric Medicine, Department of Medicine, University of Miami Miller School of Medicine, Miami, FL, USA
[3] Sylvester Comprehensive Cancer Center, University of Miami Miller School of Medicine, Miami, FL, USA

Corresponding author
Carlos Perez-Stable,
cperez@med.miami.edu

## ABSTRACT

Castration-resistant prostate cancer (CRPC) expresses high levels of the anti-apoptotic proteins Bcl-2, Bcl-xL and Mcl-1, resulting in resistance to apoptosis and association with poor prognosis. Docetaxel, an antimitotic drug that is the first-line treatment strategy for CRPC, is known to provide a small survival benefit. However, docetaxel chemotherapy alone is not enough to counteract the high levels of Bcl-2/Bcl-xL/Mcl-1 present in CRPC. ABT-737 is a small molecule that binds to Bcl-2/Bcl-xL (but not Mcl-1) with high affinity and disrupts their interaction with pro-apoptotic Bax/Bak, thus enhancing apoptosis. Our results indicate that ABT-737 can sensitize androgen-dependent LNCaP and CRPC PC3 cells to docetaxel- and to the novel antimitotic ENMD-1198-mediated caspase-dependent apoptosis. CRPC DU145 cells, however, are more resistant to ABT-737 because they are Bax null and not because they express the highest levels of anti-apoptotic Mcl-1 (associated with ABT-737 resistance). Knockdown of Bax or Bak in LNCaP indicates that ABT-737-induced antimitotic enhancement of apoptosis is more dependent on the levels of Bax than Bak. Furthermore, we find that the ability of docetaxel to increase cyclin B1/Cdk1-mediated phosphorylation of Bcl-2/Bcl-xL and decrease Mcl-1 is required for ABT-737 to enhance apoptosis in PC3 cells, as determined by addition of Cdk1 inhibitor purvalanol A and expression of shRNA specific for cyclin B1. Overall, our data suggests that the high levels of anti-apoptotic proteins in Bax-expressing CRPC cells can be overcome by targeting Bcl-2/Bcl-xL with ABT-737 and Mcl-1 with antimitotics.

## INTRODUCTION

Prostate cancer (PCa) is a leading cause of cancer-related death in men and remains incurable in the metastatic setting. Despite the initial response to androgen deprivation,

PCa gradually progresses to castration-resistant prostate cancer (CRPC) (*Hadaschik & Gleave, 2007*; *Attar, Takimoto & Gottardis, 2009*). Docetaxel (Doc) is an FDA approved first-line treatment for patients with CRPC but confers only a small survival benefit (*Tannock et al., 2004*). Once CRPC patients fail Doc chemotherapy, only the Doc derivative cabazitaxel confers a slightly longer overall survival (*de Bono et al., 2010*). To further improve overall survival of CRPC patients, a better mechanistic understanding of Doc-induced CRPC cell death is required to develop more effective combinatorial treatments.

The anti-proliferative activity of Doc results from its ability to bind microtubules and disrupt mitosis (*Jordan & Wilson, 2004*). Doc activates the mitotic checkpoint and blocks the degradation of cyclin B1, leading to a prolonged activation of cyclin-dependant kinase 1 (Cdk1) and increased mitotic arrest, followed by induction of mitotic catastrophe or apoptosis and also lysosome-dependent cell death (*Castedo et al., 2004*; *Mediavilla-Varela et al., 2009*). We have previously shown that small molecule inhibitors of Cdk1 can prevent Doc-mediated increase in cyclin B1/Cdk1 activity and block induction of apoptosis in CRPC cells (*Perez-Stable, 2006*; *Gomez, de las Pozas & Perez-Stable, 2006*). This finding indicates that prolonged cyclin B1/Cdk1 activity phosphorylates apoptotic signaling targets that can subsequently lead to apoptosis, although the precise mechanisms have been difficult to determine. However, it is likely that the mechanism involves substrates phosphorylated by cyclin B1/Cdk1.

The mitotic response to Doc shows little variation between cell types, whereas the ability to subsequently undergo apoptosis shows large variations (*Shi, Orth & Mitchison, 2008*; *Gascoigne & Taylor, 2008*). This suggests that sensitivity to Doc depends to a greater extent on cell type-specific apoptotic signaling mechanisms rather than on pathways that mediate mitotic arrest. Bcl-2, Bcl-xL, and Mcl-1 are anti-apoptotic proteins of the Bcl-2 family that are highly expressed in CRPC, resulting in resistance to apoptosis and association with poor prognosis (*Karnak & Xu, 2010*). Bcl-2, Bcl-xL, and Mcl-1 protect cells from apoptosis by binding to Bax and Bak, pro-apoptotic members of the Bcl-2 family, thereby preventing their homodimerization. Bax and Bak homodimers promote apoptosis by forming pores in the mitochondria, leading to mitochondrial outer membrane permeabilization (MOMP), cytochrome c release, and the activation of the caspase cascade (*Chipuk et al., 2010*).

Interestingly, microtubule inhibitors such as Doc induce Bcl-2 and Bcl-xL phosphorylation, thus antagonizing their anti-apoptotic function (*Haldar, Jena & Croce, 1995*; *Poruchynsky et al., 1998*). Furthermore, phospho-defective Bcl-2 and Bcl-xL mutants block the pro-apoptotic effects of microtubule inhibitors, reinforcing the notion that mitotic phosphorylation of Bcl-2 and Bcl-xL inhibits their anti-apoptotic function (*Haldar, Basu & Croce, 1998*; *Basu & Haldar, 2003*; *Upreti et al., 2008*; *Terrano, Upreti & Chambers, 2010*). Recent biochemical data now shows that cyclin B1/Cdk1 is the kinase that phosphorylates Bcl-2 and Bcl-xL during prolonged mitosis after treatment with the microtubule inhibitor vinblastine (*Terrano, Upreti & Chambers, 2010*). In addition, cyclin B1/Cdk1 can also phosphorylate Mcl-1 to increase its degradation during prolonged mitotic arrest (*Harley et al., 2010*; *Wertz et al., 2011*). During a normal mitotic cell cycle phase, cyclin B1/Cdk1 only

transiently phosphorylates Bcl-2, Bcl-xL, and Mcl-1, thus limiting the pro-cell death effect. Therefore, it is likely that during Doc-mediated mitotic arrest, prolonged cyclin B1/Cdk1 activity hyperphosphorylates Bcl-2, Bcl-xL, and Mcl-1 to block their anti-apoptotic function, which is likely important for increasing cell death.

It is known that Doc chemotherapy alone is not enough to overcome the high levels of Bcl-2, Bcl-xL, and Mcl-1 present in patients with CRPC (*Karnak & Xu, 2010*). A chemical library identified ABT-737 as a small molecule that binds Bcl-2/Bcl-xL (but not Mcl-1) with high affinity to disrupt their interaction with Bax/Bak and enhance the apoptotic signals, especially when combined with other chemotherapeutic drugs (*Oltersdorf et al., 2005*; *Tagscherer et al., 2008*). The purpose of the present study is to determine whether ABT-737 combined with Doc or a novel antimitotic ENMD-1198 (a more stable and potent derivative of 2-methoxyestradiol with clinical promise (*LaVallee et al., 2008*; *Zhou et al., 2011*)) can overcome the high levels of Bcl-2/Bcl-xL/Mcl-1 in CRPC cells and enhance apoptotic cell death. Our results show that ABT-737 enhances Doc and 1198-mediated caspase-dependent apoptosis in some PCa cells and that this enhancement is dependent on expression of Bax and on cyclin B1/Cdk1-mediated phosphorylation of Bcl-2/Bcl-xL and decrease in Mcl-1.

## MATERIALS AND METHODS

### Reagents

ABT-737 was obtained from Abbott Laboratories (Abbott Park, IL, USA), Doc from Sanofi-Aventis (Bridgewater, NJ, USA), and ENMD-1198 from EntreMed, Inc (Rockville, MD, USA). Q-VD pan-caspase inhibitor was purchased from R&D Systems (Minneapolis, MN, USA); purvalanol A from A.G. Scientific (San Diego, CA, USA); Trypan blue (0.4%) from Invitrogen (Grand Island, NY, USA); and Coomassie Blue from EMD Chemicals (Billerica, MA, USA). All other reagents were purchased from Sigma-Aldrich (St. Louis, MO, USA).

### Cell culture

Human PCa cell lines LNCaP, DU145, and PC3 were obtained from the American Type Culture Collection (Manassas, VA, USA) (*van Bokhoven et al., 2003*) and used within 6 months of resuscitation of original cultures. All cells were maintained in RPMI 1640 medium (Invitrogen) with 5% fetal bovine serum (Hyclone, Waltham, MA), 100 U/ml penicillin, 100 µg/ml streptomycin, and 0.25 µg/ml amphotericin (Invitrogen). Unlike LNCaP, LN-AI cells are able to grow for long-term in RPMI 1640 with 5% charcoal-stripped fetal bovine serum (Hyclone) and are referred to as LN-AI/CSS (*Gomez, de las Pozas & Perez-Stable, 2006*).

### Western blot analysis

Preparation of total protein lysates and Western blot analysis was done as previously described (*Gomez, de las Pozas & Perez-Stable, 2006*). The following antibodies were used: Bcl-2 (N-19), Bax (N-20), Mcl-1 (S-19), cyclin B1 (GNS1), AIF (E-1), and horseradish peroxidase-conjugated secondary antibody from Santa Cruz Biotechnology (Santa Cruz,

CA, USA); Bak (NT) from EMD Millipore (Billerica, MA, USA); Bcl-xL (#610211), cytochrome c (7H8.2C12), Smac (#612245) from BD Biosciences (San Diego, CA, USA); cleaved PARP (9541), phospho-(Ser70) Bcl-2 (2827), CoxIV (#4844), Bid (#2002) from Cell Signaling Technology (Danvers, MA, USA); phospho-(Ser62) Bcl-xl (30655) from Abcam (Cambridge, MA, USA); and Noxa (114C307.1) from Novus Biologicals (Littleton, CO, USA). After immunodetection, our preference for loading controls was for staining of total proteins transferred to the membrane with Coomassie Blue because drug treatments often affect the levels of typical housekeeping proteins such as actin or tubulin.

## ABT-737 cell viability assay

LNCaP, DU145, and PC-3 cells were seeded in 96-well plates. The next day, fresh media containing ABT-737 (1, 2.5, 5, 10 µM), or control (0.1% DMSO) were added and cells incubated for three days. The CellTiter Aqueous cell proliferation colorimetric method from Promega (Madison, WI, USA) was used to determine cell viability, as per manufacturer's instructions. Cell viability was normalized against the vehicle control and the data expressed as a percentage of control from three independent experiments done in triplicate.

## Drug treatments

PCa cells were cultured in media containing Doc (1 nM), 1198 (1 µM), ABT-737 (1 µM), Doc or 1198 + ABT-737, Q-VD (10 µM), purvalanol A (5 µM), Doc or 1198 + Q-VD or purvalanol A, or DMSO (0.1%) control for varying times (24–72 h). In all the experiments, floating and trypsinized attached cells were pooled for further analysis.

## Trypan blue exclusion assay

Treated and control PCa cells were harvested, resuspended in PBS, diluted 1:1 in 0.4% trypan blue, dead blue and live non-blue cells immediately counted using a hemacytometer, and the % dead blue cells determined from at least three independent experiments done in duplicate.

## Annexin-FITC/propidium iodide (PI) flow cytometry

Treated and control PCa cells were resuspended in binding buffer followed by the addition of annexin V-FITC and PI (Annexin V Kit sc-4252 AK; Santa Cruz Biotechnology). After 20 min, cells were analyzed by flow cytometry using a Coulter XL flow cytometer and the percentage of annexin+ cells determined using WinMDI version 2.8 from two independent experiments done in triplicate.

## Mitochondrial protein release assay

Treated and control PCa cells were resuspended in a buffer containing 100–200 µM digitonin, 20 mM Hepes, pH 7.5, 10 mM KCl, 1.5 mM MgCl, 1 mM EGTA, 1 mM EDTA, 1 mM DTT, 250 mM sucrose, and protease inhibitors (Roche, Nutley, NJ) at 50 µl/1 $\times 10^6$ cells. After 5 min on ice, cells were centrifuged at 14k rpm 5 min and the supernatant used for Western blot analysis. Digitonin is a detergent that preferentially permeabilizes plasma membrane compared to mitochondrial membrane (*Gottlieb & Granville, 2002*).

## Retrovirus transduction of DU145 and LNCaP with Bax

hBax C3-EGFP (plasmid 19741; Addgene, Cambridge, MA, USA) (*Nechushtan et al., 1999*) was digested with HindIII, blunt-ended with Klenow DNA polymerase, digested with EcoRI, and the 0.6 kb Bax insert ligated into pBABE puro plasmid (BamH1-blunt/EcoRI) using DNA ligase (New England Biolabs, Ipswich, MA, USA). Retrovirus production and infection were done by transfecting HEK293T cells (American Type Culture Collection) with pBABE/Bax or pBABE/EV (empty vector), pUMVC3, and pCVM-VSV-G with FuGene HD (Roche), addition of filtered (0.45 μM) media after 48 h to DU145 and LNCaP cells, and selection with puromycin (Invitrogen; 2 μg/ml) for 1 week. Cell death in DU145/Bax, DU145/EV, LNCaP/Bax, and LNCaP/EV cells treated with DMSO control were similar to parental cells (not shown).

## Lentiviral transduction of LNCaP, DU145, and PC3 with shRNA

The shRNA design, lentivirus production, and infection were done as previously described (*Stewart et al., 2003*). The following DNA oligonucleotides (Operon, Huntsville, AL, USA) targeting Mcl-1, Bax, Bak, and cyclin B1 were cloned into pLKO.1 lentivirus vector: shMcl-1 (M2): CCGGGCTGGAGA TTATCTCTCGGTACTC-GAGTACCGAGAGATAATCTCCAGCTTTTTG; shMcl-1 (M3): CCGGGCTAAA-CACTTGAAGACCATACTCGAGTATGGTCTTCAAGTGTTTAGC TTTTTG; shBax-1: CCGGGCCGGAACTGATCAGAACCATCTCGAGATGGTTCTGATCAGTTCCGG CTTTTTG (PC3); shBax-2: CCGGGCCTCAGGATGCGTCCACCAACTCGAGTTG-GTGGACG CATCCTGAGGCTTTTTG (LNCaP); shBak-1: CCGGTGGTACGAA-GATTCTTCAAATCTC GAGATTTGAAGAATCTTCGTACCATTTTTG (LNCaP); shBak-3: CCGGATGAGTACTTCA CCAAGATTGCTCGAGCAATCTTGGTGAAG-TACTCATTTTTTG (PC3); shCyclin B1-2: CCGG GCCAAATACCTGATGGAAC-TACTCGAGTAGTTCCATCAGGTATTTGGCTTTTTG; and shCyclin B1-3: CCGGGC-CATCCTAATTGACTGGCTACTCGAGTAGCCAGTCAATTAGGATG GCTTTTTG. The control shRNA was targeted against green fluorescent protein (GFP). For Mcl-1 knockdown in DU145/Bax and PC3/shCyclin B1 cells (puromycin resistant), shMcl-1 and shGFP oligonucleotides were cloned into pLKO.1/hygromycin plasmid and transduced cells selected with 400 μg/ml hygromycin (Invitrogen) for two weeks. Cell death in DU145/shMcl-1, DU145/shGFP, DU145/Bax/shMcl-1, DU145/Bax/shGFP, LNCaP/shBax, LNCaP/shBak, LNCaP/shGFP, PC3/shBax, PC3/shBak, PC3/shCyclin B1, PC3/shGFP, PC3/shCyclin B1/shMcl-1, and PC3/shCyclin B1/shGFP cells treated with DMSO control were similar to parental cells (not shown).

## Statistical analysis

Statistical differences between drug-treated and control PCa cells were determined by two-tailed Student's $t$-test (unequal variance) with $P < 0.05$ considered significant.

## RESULTS

### CRPC cells express high anti-apoptotic Bcl-2/Bcl-xL/Mcl-1 and low or null pro-apoptotic Bax/Bak

Since ABT-737 is a Bcl-2/Bcl-xL antagonist that should promote the pro-apoptotic function of Bax/Bak, we compared the protein levels of Bcl-2, Bcl-xL, Bax, and Bak in LNCaP, DU145, and PC3 cells. LNCaP cells are androgen-dependent, contain wild-type p53, and exhibit higher sensitivity to antimitotic-mediated apoptosis relative to DU145 and PC3, which are castration-resistant and p53 mutated or null (*van Bokhoven et al., 2003*; *Reiner et al., 2009*). As expected, DU145 and PC3 expressed higher Bcl-2/Bcl-xL when compared to LNCaP cells (Fig. 1A). Expression of Mcl-1, an anti-apoptotic member of the Bcl-2 family that is not targeted by ABT-737 and is associated with chemoresistance to ABT-737 treatment (*van Delft et al., 2006*; *Chen et al., 2007*; *Lestini et al., 2009*; *Hauck et al., 2009*; *Yecies et al., 2010*), was highest in DU145 compared to PC3 and LNCaP cells. The protein levels of pro-apoptotic Bax and Bak were lower in CRPC cells compared to LNCaP; Bax is null in DU145 (*Tang et al., 1998*). These results suggest that the high anti-apoptotic Bcl-2/Bcl-xL and low pro-apoptotic Bax/Bak protein environment present in CRPC cells may benefit from ABT-737 treatment in order to enhance apoptotic cell death. However, DU145 cells were more resistant to ABT-737 as a single agent when compared to LNCaP and PC3 cells (Fig. 1B).

### ABT-737 enhances Doc/1198-mediated apoptosis in LNCaP and PC3 but not in DU145 cells

CRPC cells such as DU145 and PC3 are more resistant to Doc treatment compared to androgen-dependent cells such as LNCaP and combinations with other drugs or agents are required to increase therapeutic efficacy. Our results showed that the combination of 1 nM Doc or 1 μM 1198 with a sub-cytotoxic dose of ABT-737 (1 μM) significantly increased cell death and cleaved-PARP (measure of caspase activity) compared to the single agents in LNCaP and PC3 but not in DU145 cells (Figs. 2A and 2B; Fig. S1A). Similar results were obtained in LNCaP-AI/CSS, a CRPC variant of LNCaP that is more chemoresistant (Fig. S1B). The pan-caspase inhibitor Q-VD (10 μM) blocked Doc+ ABT-737-mediated cell death and cleaved-PARP, indicating that increased caspase activity was required (Figs. 2A and 2B).

ABT-737 targets the mitochondria to initiate the intrinsic pathway of apoptosis by increasing the release of mitochondrial proteins such as cytochrome c, which in turn activates the caspase cascade (*Chipuk et al., 2010*). Our results indicated that ABT-737 enhanced Doc-mediated release of cytochrome c, Smac (blocks inhibitor of apoptosis [IAP] family; *LaCasse et al., 2008*), and apoptosis-inducing factor (AIF; translocates to nucleus to increase DNA fragmentation; *Susin et al., 1999*) from the mitochondria in LNCaP and PC3 but not in DU145 cells (Fig. 2C; Fig. S2). In addition, there was less cytoplasmic Bax protein in Doc + ABT-737 treated LNCaP and PC3 cells, likely as a result of greater translocation of Bax to the mitochondria. Thus, ABT-737 enhancement

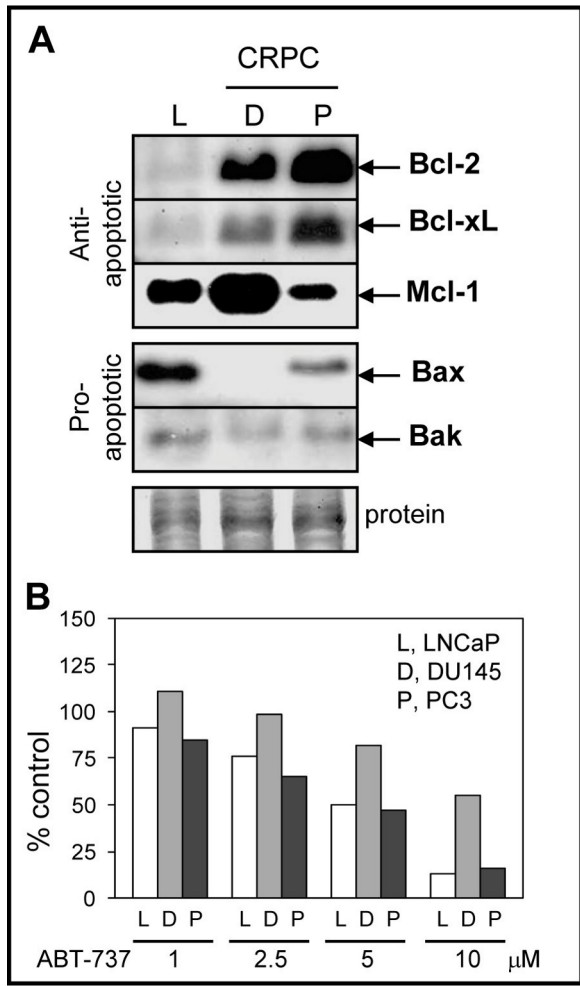

**Figure 1  Bcl-2 family protein levels and sensitivity to ABT-737 in PCa cells.** (A) Western blot analysis showing that the levels of anti-apoptotic Bcl-2 and Bcl-xL proteins are higher in DU145 (D) and PC3 (P) CRPC cells compared to androgen-dependent LNCaP (L) cells, whereas anti-apoptotic Mcl-1 is highest in DU145. The levels of pro-apoptotic Bax and Bak are higher in LNCaP compared to DU145 (Bax null) and PC3. After detection, Coomassie Blue stain of total protein transferred to the membrane is the loading control. (B) Cell viability assay showing that LNCaP and PC3 are similarly sensitive to various concentrations of ABT-737 (1–5 μM; three days), whereas DU145 is more resistant.

of Doc-mediated pro-apoptotic protein release from the mitochondria correlates with increased apoptotic cell death in LNCaP and PC3 but not in DU145.

## Bax expression but not Mcl-1 suppression sensitizes DU145 to ABT-737 enhancement of Doc/1198-mediated apoptosis

We investigated the mechanisms by which DU145 cells are more resistant to ABT-737. One possibility stems from the Bax null status of DU145 cells, which would indicate that ABT-737 mediates its cytotoxicity via the Bax pathway. This is supported by a previous finding indicating that transient transfection of Bax into DU145 cells increases sensitivity to ABT-737 + TRAIL (*Song, Kandasamy & Kraft, 2008*). Another possibility

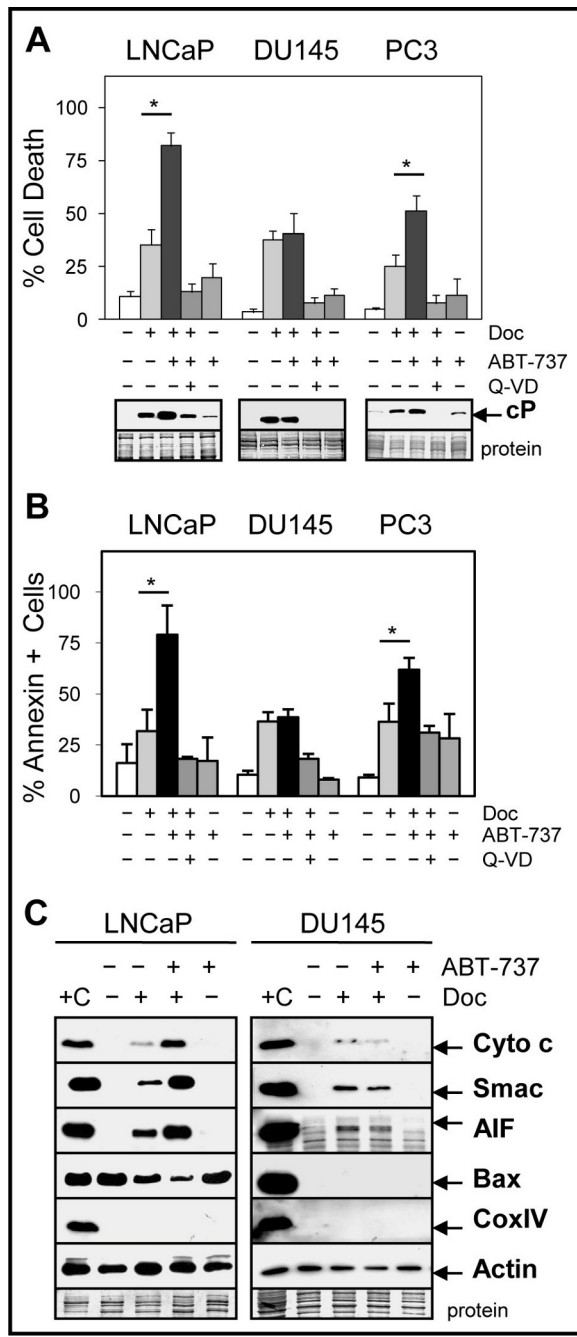

**Figure 2 ABT-737 enhances Doc-mediated apoptotic cell death in LNCaP and PC3 but not in DU145 PCa cells.** (A) Trypan blue exclusion assay showing that the combination of 1 nM Doc + 1 μM ABT-737 increases total cell death in LNCaP and PC3 but not in DU145 compared to Doc or ABT-737 alone (*, $P < 2 \times 10^{-4}$). LNCaP were treated for 48 h and DU145 and PC3 for 72 h. Pan-caspase inhibitor Q-VD (10 μM) blocks the Doc + ABT-737 increase in cell death. Western blot analysis showing that Doc + ABT-737 increases cleaved-PARP (cP) levels in LNCaP and PC3 but not in DU145 compared to Doc or ABT-737 alone. Q-VD blocks the Doc + ABT-737 increase in cP in all cells. (B) Flow cytometric analysis showing higher annexin-FITC stained LNCaP and PC3 but not (continued on next page...)

**Figure 2 (...continued)**
DU145 cells treated with Doc + ABT-737 compared to Doc or ABT-737 alone (*, $P < 2 \times 10^{-5}$). Q-VD blocks the Doc + ABT-737 increase in annexin + cells in LNCaP and PC3. (C) Mitochondrial protein release assay and Western blot showing increased cytochrome $c$, Smac, AIF and decreased Bax in LNCaP cells treated with Doc + ABT-737 compared to Doc or ABT-737 alone, and control. In DU145 cells, cytochrome c, Smac, and AIF mitochondrial release were similar in Doc + ABT-737 as in Doc alone. Cox IV protein is negative indicating no mitochondrial contamination whereas actin is the positive control. + C for both LNCaP and DU145 is the lysate prepared from LNCaP cells using the standard method for total proteins.

is that ABT-737 resistance of DU145 cells arises from elevated Mcl-1, which unlike Bcl-2/Bcl-xL does not interact with ABT-737 and may therefore block the ability of ABT-737 to increase apoptosis (*van Delft et al., 2006*; *Chen et al., 2007*; *Lestini et al., 2009*; *Hauck et al., 2009*; *Yecies et al., 2010*). To address these possibilities, we isolated DU145 cells stably expressing Bax and DU145 cells with stable Mcl-1 knockdown. Our results showed that ABT-737 significantly enhanced Doc/1198-mediated cell death in DU145/Bax cells above Doc/1198 treatment alone (Figs. 3A and 3B). Similar results were obtained in LNCaP/Bax cells (Fig. S3). In contrast, knockdown of Mcl-1 in DU145 or DU145/Bax cells did not significantly enhance Doc/1198 + ABT-737-mediated apoptotic cell death (Fig. 3C; Fig. S4). These results indicate that DU145 cells are more resistant to ABT-737 enhancement of antimitotic-mediated apoptosis because they are Bax null and not because they express high Mcl-1. In contrast, knockdown of Mcl-1 in LNCaP and PC3 cells increased cell death and cl-PARP, confirming the importance of Mcl-1 in resistance to Doc/1198 + ABT-737 treatment (Fig. S5).

## ABT-737-mediated enhancement of Doc/1198-induced apoptosis in LNCaP is more dependent on Bax than Bak

ABT-737 inhibits the interactions of Bcl-2/Bax and Bcl-xL/Bak, thus allowing Bax and Bak to induce MOMP and apoptosis (*Oltersdorf et al., 2005*; *Tagscherer et al., 2008*). To further investigate the relative importance of Bax versus Bak in mediating ABT-737 enhancement of Doc/1198-mediated cell death in PCa cells, we isolated LNCaP and PC3 cells stably expressing either shBax or shBak. Results showed that knockdown of Bax lowered Doc/1198 + ABT-737-mediated cell death and cleaved-PARP in both LNCaP and PC3 cells compared to the negative control shGFP cells (Fig. 4A). Knockdown of Bak also lowered Doc/1198 + ABT-737-induced cell death and cleaved-PARP in PC3 cells but had no significant effects in LNCaP cells (Fig. 4B). These results suggest that ABT-737 enhancement of Doc/1198-mediated apoptosis in LNCaP cells is more dependent on Bax than to Bak but PC3 cells are dependent on both Bax and Bak.

## Doc counteracts the ABT-737-mediated increase in Mcl-1

We examined whether Doc and Doc + ABT-737 had any effects on the protein levels of Bcl-2 family members in LNCaP and PC3 cells. Interestingly, Doc decreased and ABT-737 increased Mcl-1 but the combination of Doc + ABT-737 lowered Mcl-1 (Fig. 5). The mechanism by which ABT-737 alone increases Mcl-1 protein is not known but may reflect

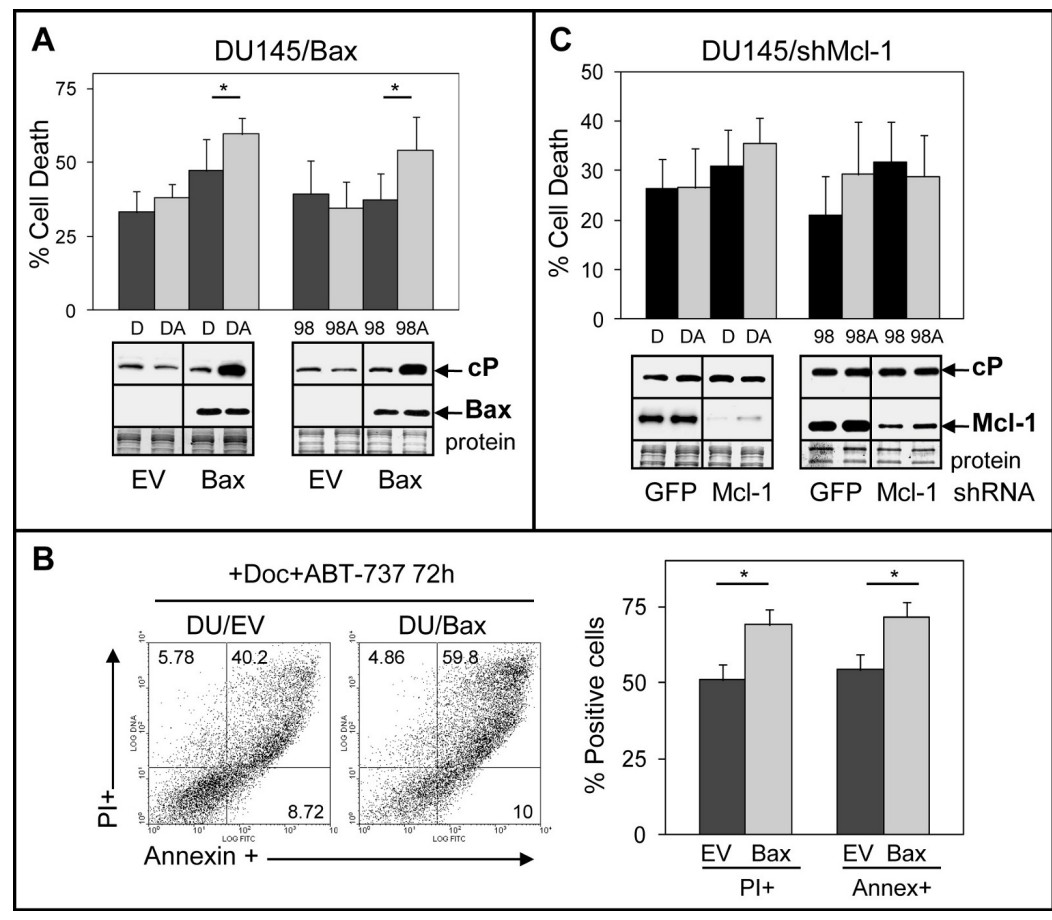

**Figure 3 DU145 cells are resistant to ABT-737 because they are Bax null.** (A) Trypan blue exclusion assay showing that Doc + ABT-737 (DA) or 1198 + ABT-737 (98A) increases cell death at 72 h in DU145/Bax compared to Doc (D) or 1198 (98) alone (*, $P < 0.008$), whereas there is no effect in DU145/EV (empty vector) control cells. Western blot analysis showing that DA or 98A increases cP in DU145/Bax cells compared to D or 98 alone, whereas there is no increase of cP in DU/EV cells. Bax is expressed in DU145/Bax but not in DU145/EV cells. (B) Annexin-FITC/PI flow cytometric analysis showing higher annexin+ and PI+ cells in DA treated DU145/Bax compared to DU145/EV cells (*, $P < 1 \times 10^{-6}$). (C) Trypan blue exclusion assay showing that DA or 98A does not significantly increase cell death in DU145/shMcl-1 and DU145/shGFP control cells compared to D or 98 alone. Western blot analysis showing no difference in the cP levels in DU145/shMcl-1 and DU145/shGFP control cells treated with D, DA, 98, or 98A. Mcl-1 is expressed much higher in DU145/shGFP compared to DU145/shMcl-1 cells.

the observation that acquired resistance to ABT-737 involves increased Mcl-1 (*Yecies et al., 2010*). There were no clear differences in the protein levels of Bcl-2, Bcl-xL, Bax, Bak, Bid, and Noxa with the exception in LNCaP where there was less Bcl-2 (Doc, 24 h) and Bak (Doc + ABT-737, 48 h). These results suggest that the Doc + ABT-737 enhancement of apoptosis may depend upon the ability of Doc to counteract the ABT-737-mediated increase in Mcl-1.

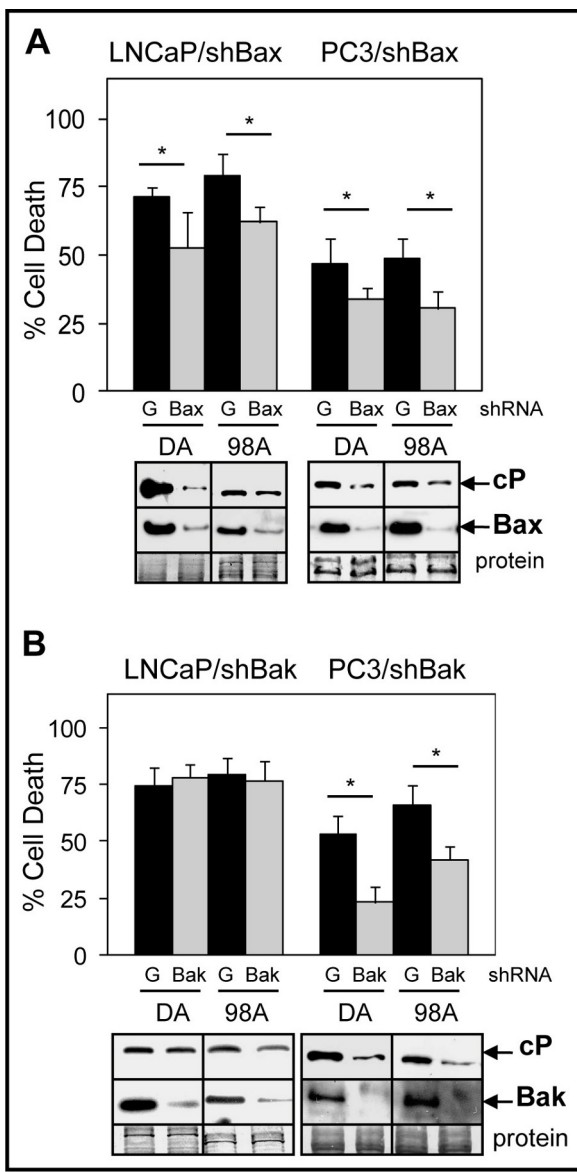

**Figure 4 Bax suppression has a greater effect on ABT-737-mediated Doc/1198-induced apoptotic cell death than Bak suppression in LNCaP cells.** (A) Trypan blue exclusion assay showing significantly less cell death in LNCaP/shBax and PC3/shBax cells treated with DA or 98A compared to shGFP (G) control cells (*, $P < 0.02$). Western blot analysis showing less cP and Bax in LNCaP/shBax and PC3/shBax cells treated with DA or 98A compared to control cells. (B) Trypan blue exclusion assay showing less cell death in PC3/shBak (*, $P < 6 \times 10^{-4}$) but not in LNCaP/shBak cells treated with DA or 98A compared to control cells. Western blot analysis showing less cP in PC3/shBak cells treated with D or 98 + A but little difference in LNCaP/shBak cells compared to control cells. Bak is lower in LNCaP/shBak and PC3/shBak cells compared to control cells.

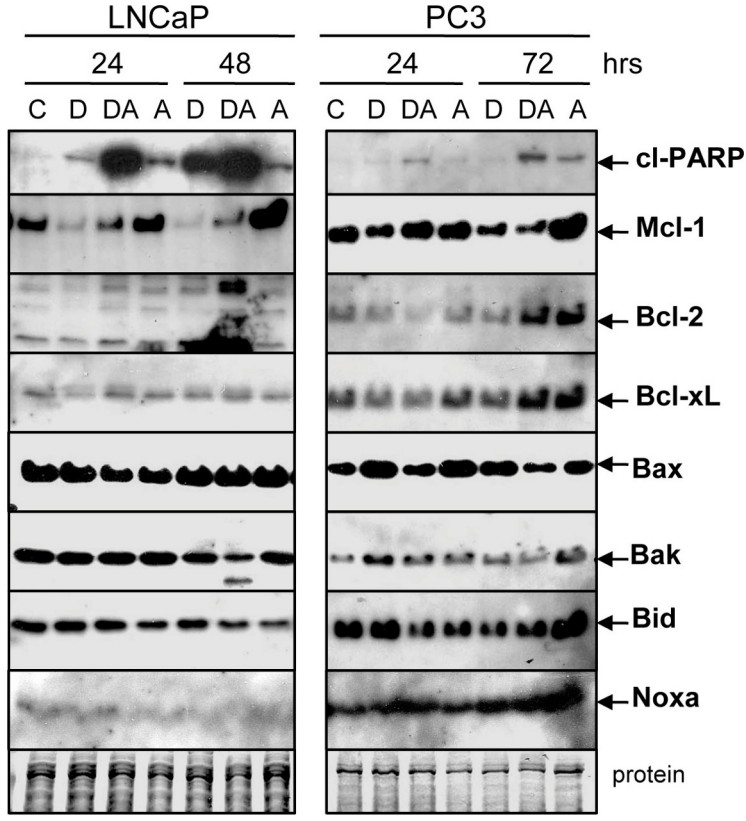

**Figure 5 Doc counteracts the ABT-737-mediated increase in Mcl-1 protein.** Western blot showing that treatment of LNCaP and PC3 cells with 1 µM ABT-737 (A) increases Mcl-1 but treatment with 1 nM Doc (D) decreases Mcl-1. Combination of Doc + ABT-737 (DA) decreases Mcl-1. In LNCaP, there is less Bcl-2 (D, 24 h) and Bak (DA, 48 h) but few differences in Bcl-xL, Bax, Bid, and Noxa. In PC3, there are few differences in Bcl-2, Bcl-xL, Bax, Bak, Bid, and Noxa.

## ABT-737-mediated enhancement of Doc-induced apoptosis is dependent on cyclinB1/Cdk1-mediated phosphorylation of Bcl-2/Bcl-xL and decrease of Mcl-1

Small molecule inhibitors of Cdk1 prevent Doc-mediated increase in cyclin B1/Cdk1 activity and blocks induction of apoptosis in CRPC cells (*Perez-Stable, 2006*; *Gomez, de las Pozas & Perez-Stable, 2006*). We investigated whether cyclin B1/Cdk1-mediated increase in Bcl-2/Bcl-xL phosphorylation and decrease in Mcl-1 is important for the ABT-737 enhancement of Doc-induced apoptosis. Treatment (Doc alone and Doc + ABT-737) of LNCaP and PC3 but not DU145 cells increased phospho (P)-Bcl-2, whereas the levels of P-Bcl-xL was similar in all three cell lines (Fig. S6). Addition of 5 µM purvalanol A, a specific inhibitor of cyclin B1/Cdk1 activity (*Gray et al., 1998*) blocked Doc + ABT-737 cell death and the increase in P-Bcl-2/P-Bcl-xL in PC3 cells (Fig. 6A). However, purvalanol A increased Mcl-1 protein in Doc + ABT-737 treated PC3 cells, suggesting that this may also play an important role in blocking apoptotic cell death (Fig. 6A).

To further determine whether the Doc-mediated increase in cyclin B1/Cdk1 activity, P-Bcl-2/P-Bcl-xL, and decrease in Mcl-1 is important for ABT-737 + Doc cell death,

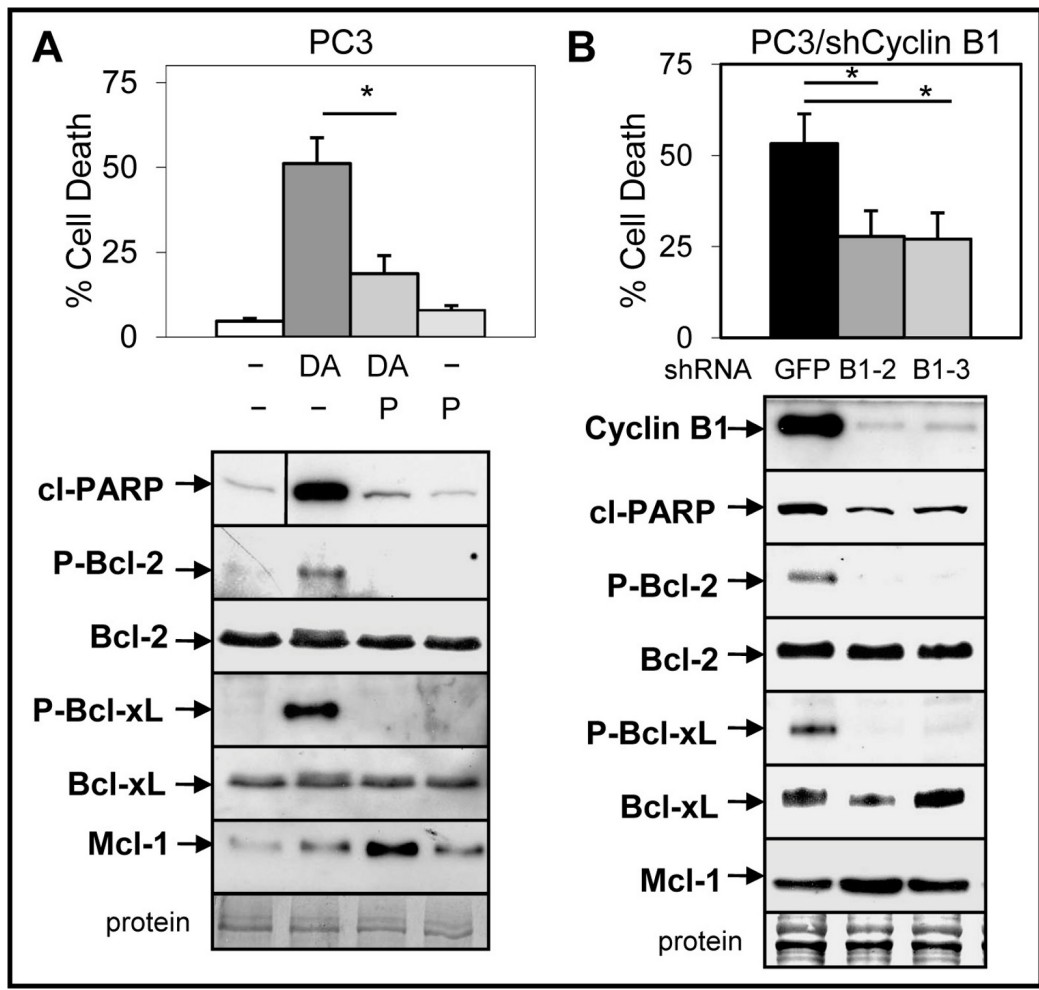

**Figure 6 Inhibition of cyclinB1/Cdk1-mediated phosphorylation of Bcl-2/Bcl-xL blocks ABT-737 enhancement of Doc induced apoptosis in PC3 cells.** (A) Trypan blue exclusion assay showing that 5 µM purvalanol A (P) lowers DA cell death in PC3 cells (*, $P < 9 \times 10^{-8}$). Western blot analysis showing that P blocks the DA increase in cl-PARP, P-Bcl-2 and P-Bcl-xL, whereas there is no change in the total levels of Bcl-2 and Bcl-xL; P treatment increases Mcl-1 in DA treated cells. Vertical line in cl-PARP indicates sample from same blot not in sequence. (B) Trypan blue exclusion assay showing less cell death in DA treated PC3/shCyclin B1-2 and -3 cells compared to PC3/shGFP control cells (*, $P < 4 \times 10^{-5}$). Western blot analysis showing less cyclin B1, cP, P-Bcl-2, P-Bcl-xL, and slightly greater Mcl-1 in PC3/shCyclin B1-2 and -3 cells treated with DA compared to PC3/shGFP control cells.

we isolated PC3 cells stably expressing shCyclin B1. Results showed that knockdown of cyclin B1 lowered Doc + ABT-737 cell death, cleaved-PARP, and P-Bcl-2/P-Bcl-xL in PC3/shCyclin B1 cells compared to shGFP control cells (Fig. 6B). However, given that there is less cyclin B1 to mediate Cdk1 degradation of Mcl-1 by Doc, it is not surprising that Mcl-1 levels are slightly higher in Doc + ABT-737 treated PC3/shCyclin B1 compared to shGFP cells (Fig. 6B). We then isolated PC3/shCyclin B1 cells stably expressing shMcl-1 and the results showed that Doc + ABT-737 increased cell death (67%) nearly to the levels of PC3/shMcl-1 cells (83%) (Fig. 7; Fig. S5B). Overall, these results suggest that the

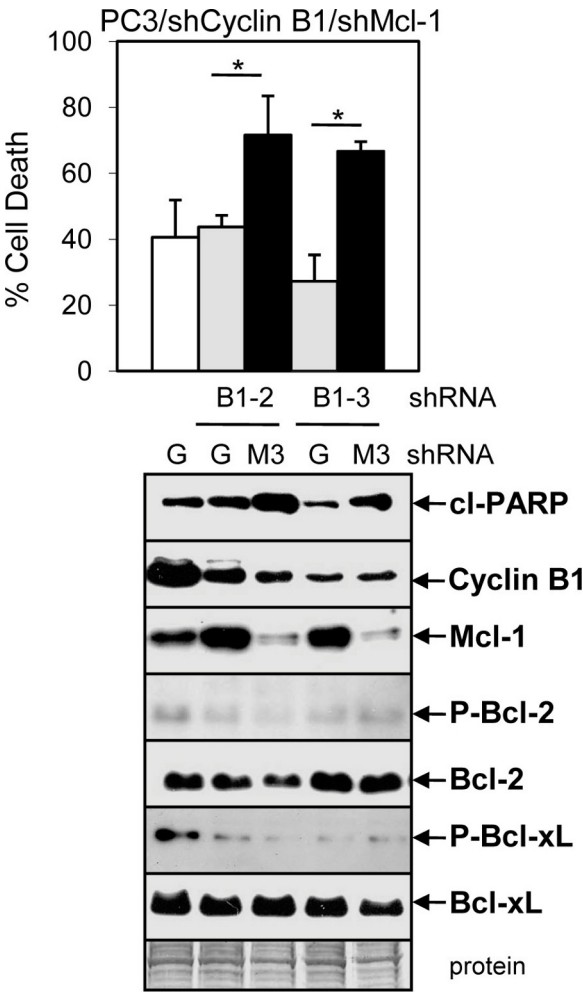

**Figure 7 ABT-737 enhancement of Doc-mediated apoptosis is more dependent on the ability of cyclin B1/Cdk1 to decrease Mcl-1 than to phosphorylate Bcl-2/Bcl-xL.** Trypan blue exclusion assay showing greater cell death in DA treated (48 h) PC3/shCyclin B1/shMcl-1 (B1-2/M3, B1-3/M3) cells compared to control PC3/shCyclin B1/shGFP (B1-2/G, B1-3/G) and PC3/shGFP (G) cells (*, $P < 0.002$). Western blot analysis showing increased cl-PARP and decreased Mcl-1 in PC3/shCyclin B1/shMcl-1 compared control shGFP cells after DA treatment (24 h). There is less cyclin B1, P-Bcl-1, and P-Bcl-xL in PC3/shCyclin B1/shGFP or shMcl-1 compared to PC3/shGFP cells, whereas there are no changes in total Bcl-1 or Bcl-xL.

ABT-737 enhancement of Doc-mediated apoptosis is more dependent on the ability of cyclin B1/Cdk1 to increase the degradation of Mcl-1 than to phosphorylate Bcl-2/Bcl-xL.

## DISCUSSION

Progression of PCa to CRPC is often associated with overexpression of the anti-apoptotic proteins Bcl-2, Bcl-xL, and Mcl-1, resulting in resistance to apoptosis and poor prognosis (*Karnak & Xu, 2010*). Doc is an antimitotic drug approved for the treatment of CRPC but the high levels of Bcl-2/Bcl-xL/Mcl-1 confers a block, resulting in less apoptosis and reduced efficacy. Here we report that the Bcl-2/Bcl-xL small molecule antagonist ABT-737 can overcome this block and increase Doc and 1198 (new antimitotic)-induced

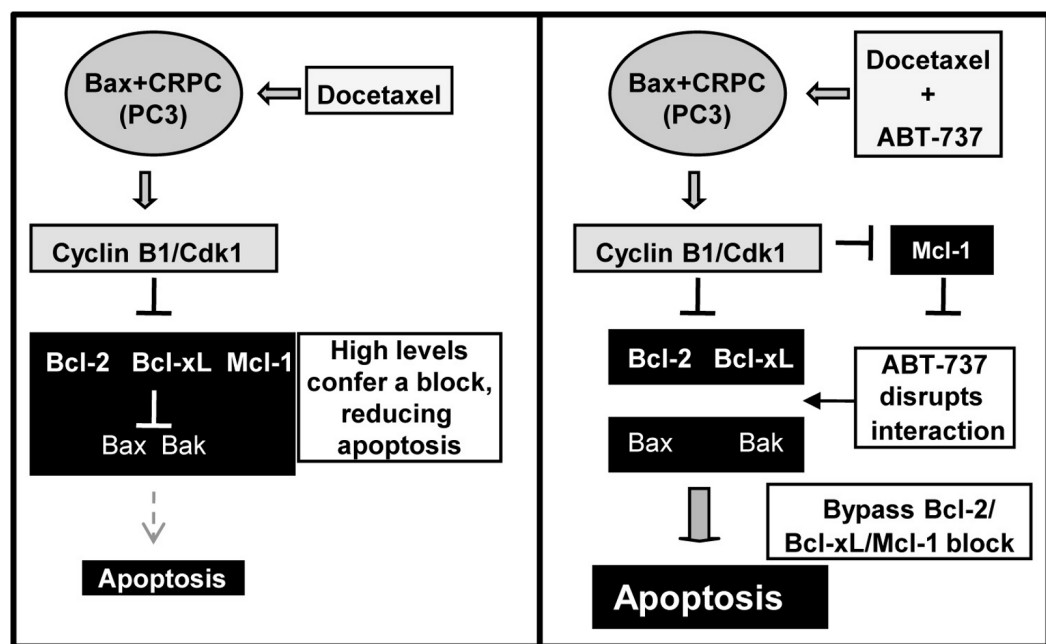

**Figure 8 Schematic of how ABT-737 can sensitize Bax+ CRPC cells to Doc.** Treatment of CRPC cells such as PC3 with Doc deregulates cyclin B1/Cdk1 activity and increases phosphorylation of Bcl-2/Bcl-xL and degradation of Mcl-1. However, the high levels of Bcl-2/Bcl-xL/Mcl-1 in CRPC cells presents a block to apoptosis. Addition of ABT-737 disrupts the anti-apoptotic activity of Bcl-2/Bcl-xL and more effectively allows Doc treatment to bypass the block and increase apoptosis. Cyclin B1/Cdk1 hyperactivity caused by Doc treatment also lowers Mcl-1, an ABT-737 resistance factor, to further increase apoptosis.

apoptosis in CRPC PC3 and LN-AI/CSS cells. However, another CRPC cell line, DU145, is more resistant to ABT-737 due to the lack of the pro-apoptotic Bax protein and the combination with Doc or 1198 did not further increase apoptosis. Our results also indicate that the ABT-737 enhancement of Doc-mediated apoptosis in LNCaP and PC3 cells is dependent upon the ability of Doc to constitutively activate cyclin B1/Cdk1 activity and hyperphosphorylate Bcl-2/Bcl-xL and decrease Mcl-1. Overall, these results provide mechanistic insight into how ABT-737 can sensitize Bax expressing CRPC cells to Doc/1198 treatment by overcoming the block in apoptosis due to the high levels of Bcl-2/Bcl-xL/Mcl-1 (Fig. 8).

There is strong evidence that Bcl-2/Bcl-xL overexpression is important for CRPC progression (*Miyake, Monia & Gleave, 2000*). Our results indicate that the commonly utilized CRPC cell lines DU145 and PC3 express higher levels of Bcl2/Bcl-xL and lower levels of Bax/Bak when compared to androgen-dependent LNCaP cells (Fig. 1A). Several reports indicate that the efficacy of ABT-737 positively correlates with Bcl-2 levels, i.e., the higher the Bcl-2 the better the response to ABT-737 alone or in combination with other chemotherapeutic agents (*Del Gaizo Moore et al., 2008*; *Hann et al., 2008*; *Mason et al., 2009*; *Oakes et al., 2012*). This correlation does not appear to apply to PCa cell lines as LNCaP has the lowest levels of Bcl-2 yet are as sensitive to ABT-737 as PC3 cells with the highest level of Bcl-2 (Fig. 1).

Numerous reports indicate that Mcl-1 overexpression in a variety of cancers can mediate resistance to ABT-737 (*van Delft et al., 2006*; *Chen et al., 2007*; *Lestini et al., 2009*; *Hauck et al., 2009*; *Yecies et al., 2010*). However, our results indicate that DU145 cells are more resistant to ABT-737 as a single agent and in combination with Doc/1198 due to the lack of Bax expression and not because they express the highest levels of Mcl-1. In contrast, knockdown of Mcl-1 in Bax expressing LNCaP and PC3 cells enhances sensitivity to Doc/1198 + ABT-737. It is possible that, in PCa cells, the complete loss of Bax (as in DU145) is a more dominant mechanism for ABT-737 resistance compared to overexpression of Mcl-1. In PC3, however, the Bax levels are much lower than in LNCaP, yet both cells respond similarly to ABT-737, suggesting that only a low amount of Bax protein is required. In PCa clinical biopsies, loss of Bax immunostaining relative to normal non-cancer prostate epithelium is a useful biomarker for categorizing patient risk and response to radiation therapy (*Pollack et al., 2003*; *Khor et al., 2007*). Therefore, determining the percentage of Bax negative cells in human PCa specimens may provide a useful biomarker for identifying patients that should respond to ABT-737 treatment.

During a normal mitotic cell cycle phase, cyclin B1/Cdk1 transiently phosphorylates numerous substrates, including the anti-apoptotic proteins Bcl-2, Bcl-xL, and Mcl-1. Antimitotic drugs such as Doc or 1198 can prevent the degradation of cyclin B1, resulting in constitutively active Cdk1 activity and hyperphosphorylation of Bcl-2/Bcl-xL/Mcl-1. Our results suggest that cyclinB1/Cdk1-mediated phosphorylation of Bcl-2/Bcl-xL is important for the enhancement of apoptosis by the Doc + ABT-737 combination in LNCaP and PC3 cells. This supports previous results that the antimitotic drug vinblastine promotes apoptosis in HeLa cervical carcinoma cells by a similar mechanism (*Terrano, Upreti & Chambers, 2010*). More recently, the sensitivity of cancer cells to antimitotic drugs has been shown to have a dependence on the ability of cyclin B1/Cdk1 to phosphorylate and increase the degradation of Mcl-1 in order to enhance apoptosis (*Harley et al., 2010*; *Wertz et al., 2011*). Our results suggest that the antimitotic-mediated increase in cyclin B1/Cdk1 activity and reduction of Mcl-1 protein counteracts the increase of Mcl-1 by ABT-737, resulting in enhanced apoptosis. Antimitotic drugs or radiation treatment also enhance cyclin B1/Cdk1 phosphorylation of numerous other substrates that are implicated in either increasing or decreasing apoptosis (*O'Connor et al., 2000*; *Konishi et al., 2002*; *Berndtsson et al., 2005*; *Allan & Clarke, 2007*; *Andersen et al., 2009*; *Nantajit et al., 2010*). Overall, our results suggest that the cyclin B1/Cdk1-mediated hyperphosphorylation of Bcl-2, Bcl-xL, and Mcl-1 is a major mechanism linking mitotic arrest to the induction of apoptosis in PCa cells.

Although our results do not address whether the Doc/1198 + ABT-737 combination will be effective in vivo, there is evidence in leukemia, and lung, prostate and breast cancer to indicate that the antimitotic + ABT-737 combination should prove to be effective in animal models of PCa (*Oakes et al., 2012*; *Shoemaker et al., 2006*; *Kang et al., 2007*; *Bray et al., 2009*). Navitoclax, previously known as ABT-263, is an orally bioavailable analog of ABT-737 with identical function that is currently in Phase II trials for refractory lymphoid malignancies and solid tumors and appears to be a promising agent for use in combination

with Doc (*Tse et al., 2008*; *Shi et al., 2011*). Our results provide a strong mechanistic rationale for combining targeted chemotherapy against Bcl2/Bcl-xL, as with navitoclax, with the currently approved drug for CRPC Doc, which targets the destruction of Mcl-1, a navitoclax resistance factor. With the addition of cabazitaxel to help in the treatment of patients that develop Doc-resistance, it is also likely that the combination of cabazitaxel and navitoclax will further improve overall survival (*de Bono et al., 2010*).

## ACKNOWLEDGEMENTS

We thank Dr. Pryamvada Rai for critical reading of the manuscript and helpful suggestions; Deanna Palenzuela for technical assistance; Ron Hamelik for assistance with flow cytometry; and Drs. Bernard Roos and Guy Howard for support.

### Funding

Carlos Perez-Stable received a grant from Veterans Affairs Merit Review (086906). The funder had no role in study design, data collection and analysis, decision to publish, or preparation of the manuscript.

### Grant Disclosures

The following grant information was disclosed by the authors:
Veterans Affairs Merit Review: 086906.

### Competing Interests

The authors declare that they have no conflict of interest.

### Author Contributions

- Ricardo Parrondo conceived and designed the experiments, performed the experiments, analyzed the data, wrote the paper.
- Alicia de las Pozas and Teresita Reiner performed the experiments.
- Carlos Perez-Stable conceived and designed the experiments, performed the experiments, analyzed the data, wrote the paper.

### Supplemental Information

Supplemental information for this article can be found online at http://dx.doi.org/10.7717/peerj.144.

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
