# Peer review of "ABT-737, a small molecule Bcl-2/Bcl-xL antagonist, increases antimitotic-mediated apoptosis in human prostate cancer cells"

_PeerJ, doi:10.7717/peerj.144_

## Round 0.1 · original submission · Minor Revisions

The manuscript that you submitted is prepared very well, with a clear, succinct and consistently styled presentation, appropriate discourse, details and references, and suitably composed figures.

Of the three reviewers, only one has suggested a few changes, all of which are minor. The comment about mentioning the phenomenon of caspase-independent cytotoxic effect of docetaxel on prostate cancer cells is relatively important and should be addressed. Regarding the comments asking whether Coomassie stain was used on protein blots or gels, and whether the LNCaP and PC3 cells that were used in the study were parental or derived, please clarify appropriately in the manuscript as other readers of the publication may have the same questions. You can choose to address the remaining comments of the reviewer at your discretion.

Reviewer 1 ·

Basic reporting

The majority of the manuscript is coherent, simplistic and intriguing.
However, the 4th paragraph of the introduction could use a little more clarification and purpose. Lines 317-320 would fit well into this paragraph.

It needs to be mentioned in the introduction or Discussion section that docetaxel induced prostate cancer death involves concomintant activation of both caspase- and lysosome-dependent pathways (Mediavilla-Varela M. et al., Mol. Cancer. 2009)

Experimental design

It is stated that ABT-737 can sensitize androgen-dependent LNCAP and PC-3 cells to docetaxel and 1198. Are both these cell lines docetaxel-resistant versions of the parental cells or are they docetaxel sensitive LNCAP and PC-3 cells?

Materials and Methods- ATCC recommends growing PC-3 cells in medium with 10% FBS.

If selection of a loading control is difficult, it is better to use Ponceaus staining of the membrane rather than Coomassie blue staining of the gel because it is the same membrane and is ideal for monitoring transfer. However, some researchers are using a new technique of protein staining with Coomassie that can be used on the membrane after immunodetection (Welinder and Ekblad, J. Proteome Res. 2011). Please clarify if you stained the gel or membrane.

Validity of the findings

No comments

Additional comments

I would suggest uniformity in the labeling of figures such as Figures 1A and 1B. I would choose between LN or L for LNCaP and DU or D for DU145. Also, I would not use PC in labeling for PC3 cells as PC has been consistently used in the rest of the paper as an acronym for prostate cancer. I would suggest identifying prostate cancer as PCa to avoid this confusion.

Reviewer 2 ·

Basic reporting

No comments.

Experimental design

This was actually very well done; the results of this manuscript appear to be the collaborative work of several different projects, which, as described in the methods section of this paper, appear to be technically sound.

Validity of the findings

The results of the author's experiments is clearly conveyed in the context of what is known in the general literature. Their final conclusions appear to be supported by their findings in the results.

Additional comments

This was an interesting investigation into some of the biologic mechanisms for chemotherapeutic resistance in the setting of Castration resistant prostate cancer with well-designed experiments to quantify the relative importance of roles played by certain pro/anti-apoptotic proteins. It appears that the scientific methods discussed and conclusions derived from this set of studies were well-designed and appropriate, respectively.

Reviewer 3 ·

Basic reporting

The article is well written and conforms to the PeerJ policies.

Experimental design

The experimental design is appropriate for the authors to make their conclusions.

Validity of the findings

The data is sound albeit not incredibly novel or surprising. I reviewed this manuscript for a different journal and that was the main conclusion of the reviews. Thus if impact is not a factor in the decision, I believe this is an acceptable paper.

Additional comments

This is an interesting finding. The experiments are well-performed, however the findings and conclusions are not terribly surprising. So the study is not incredibly novel, but it is solid and appropriate.

---

## Round 0.2 · accepted · Accept

The issues raised in the last round of review have bee adequately addressed.